:ᗴ: PLOS | ONE

# Visual body form and orientation cues do not modulate visuo-tactile temporal integration

Sophie Smit[ID][1]*, Anina N. Rich[1,2], Regine Zopf[1,3]

**1** Perception in Action Research Centre & Department of Cognitive Science, Faculty of Human Sciences, Macquarie University, Sydney, Australia, **2** Centre for Elite Performance, Expertise & Training, Macquarie University, Sydney, Australia, **3** Body Image and Ingestion Group, Macquarie University, Sydney, Australia

* sophie.smit@students.mq.edu.au

## Abstract

Body ownership relies on spatiotemporal correlations between multisensory signals and visual cues specifying oneself such as body form and orientation. The mechanism for the integration of bodily signals remains unclear. One approach to model multisensory integration that has been influential in the multisensory literature is Bayesian causal inference. This specifies that the brain integrates spatial and temporal signals coming from different modalities when it infers a common cause for inputs. As an example, the rubber hand illusion shows that visual form and orientation cues can promote the inference of a common cause (one's body) leading to *spatial integration* shown by a proprioceptive drift of the perceived location of the real hand towards the rubber hand. Recent studies investigating the effect of visual cues on *temporal integration*, however, have led to conflicting findings. These could be due to task differences, variation in ecological validity of stimuli and/or small samples. In this pre-registered study, we investigated the influence of visual information on temporal integration using a visuo-tactile temporal order judgement task with realistic stimuli and a sufficiently large sample determined by Bayesian analysis. Participants viewed videos of a touch being applied to plausible or implausible visual stimuli for one's hand (hand oriented plausibly, hand rotated 180 degrees, or a sponge) while also being touched at varying stimulus onset asynchronies. Participants judged which stimulus came first: viewed or felt touch. Results show that visual cues do not modulate visuo-tactile temporal order judgements. This is not in line with the idea that bodily signals indicating oneself influence the integration of multisensory signals in the temporal domain. The current study emphasises the importance of rigour in our methodologies and analyses to advance the understanding of how properties of multisensory events affect the encoding of temporal information in the brain.

## Introduction

Perception of one's body relies on multisensory integration and is dynamically updated based on the available sensory input [1]. Information regarding spatiotemporal correlations between inputs from the different senses is particularly important for self-recognition [2, 3]. For example, experimental work with the rubber hand illusion (RHI) shows that synchronously viewing

**Data Availability Statement:** All materials, including the presentation script and videos, raw and summary data and analysis scripts, are available at (osf.io/grw57, registration date 06/06/2018).

**Funding:** This work is supported by a Discovery Early Career Research Award from the Australian Research Council (ARC; DE140100499) to RZ. ANR has research funding from the ARC (DP12102835 and DP170101840). The funders had no role in study design, data collection and analysis, decision to publish, or preparation of the manuscript.

**Competing interests:** The authors have declared that no competing interests exist.

and feeling touch can cause signals from different modalities to become integrated, resulting in an illusionary experience of ownership over an artificial object [4–7]. Furthermore, visual cues such as form and orientation provide crucial information about whether the observed object is plausible for one's body, which can facilitate or inhibit the emergence of body ownership [6]. For instance, the RHI is reduced when the object lacks certain hand-like features [8], or when it is rotated at an improbable angle in relation to the participant's own body [9, 10]. This demonstrates that besides temporal cues, visual form and orientation cues play an important role in perceiving one's own body. However, the mechanism by which visual cues might modulate the integration of multisensory bodily stimuli still remains unclear. Further research into this process is fundamental for our understanding of how we perceive our bodies and interact with objects in the world around us. Motivated by the RHI literature, in this study we use videos of touch combined with a felt touch to investigate if form and orientation cues directly influence the temporal integration of visual and tactile inputs.

The brain constantly receives signals from different sensory modalities with some variability between the exact timing or location, and these are either combined into the same multisensory event or kept separate [11, 12]. One proposal is that the processing of multisensory signals relies on computational mechanisms for causal inference to determine the probability that the individual unisensory signals belong to the same object or event. This account holds that the brain computes probabilities for common and separate causes, which then provide the weights given to the integrated and separated perceptual estimates. These relative weights determine the degree of integration versus separation of multisensory signals [13]. Bayesian causal inference models can therefore provide a unified theory for the perception of multisensory events, including their spatial and temporal characteristics [14–18]. On this view, the degree of integration versus separation can be influenced by previous knowledge that signals belong to one and the same object or event, and repeated experience that signals are statistically likely to co-occur [19–21]. Signals that are likely to 'belong together' (also referred to as the 'assumption of unity'; Welch and Warren [22]) can become partially or completely integrated, which limits access to spatiotemporal conflict between the original signals [23]. These processes of causal inference and multisensory integration explain how the brain perceives signals that are estimated to originate from a common source as the same multisensory event.

A causal inference process might also govern the binding and integration of bodily signals [24]. Depending on whether the brain infers a common cause for inputs or not, it integrates or segregates spatial and temporal signals coming from visual, tactile and proprioceptive modalities. Visual cues such as body form and orientation could function as causal binding factors (i.e., influence the relative probabilities for a common versus a separate cause) as these indicate whether or not inputs originated from the same source (e.g., one's own hand). This could explain why the RHI typically only arises in a plausible context for one's own body and why form and orientation cues can modulate a shift in the perceived location of the real hand towards the spatially separated rubber hand [6, 25]. This 'proprioceptive drift' indicates *spatial integration* of visual information originating from the artificial hand and proprioceptive information about the location of one's own hand [6]. Further, Shimada et al. [5] showed that participants still experienced a strong illusion of ownership even when the visual stimulation on the rubber hand and tactile stimulation on the participant's own hand were presented with a 300 ms delay. This might indicate that the visual presentation of a hand enhances *temporal integration* of visual and tactile stimuli as well. However, evidence for the effect of causal inference on temporal integration of body-related signals is mixed.

When we feel a touch to our hand, we expect to also see a touch to an object that looks like a hand as opposed to a non-hand object. We furthermore observe the touch to a hand located in a set orientation relative to the arm and body, making it clear that it is our hand that is

being touched. Hence, the experience of feeling a touch to our hand is most likely to occur together with a visual touch to a plausibly oriented human hand (i.e., one's own hand), indicating that the visual and tactile touch event share a common cause. Ide and Hidaka [26] tested if form and orientation manipulations affect visuo-tactile temporal integration using simple hand images. A light flash on the left index finger of a line drawing of a forward-facing hand, an inverted hand or an arrow (the visual stimulus) was presented with different SOAs relative to a vibration to the tip of a participant's left index finger (the tactile stimulus). Participants made unspeeded temporal order judgements (TOJ) about whether the visual or the tactile stimulus was presented first and the authors measured the 'just noticeable difference' (JND, the smallest temporal interval at which participants can still reliably distinguish the temporal order of stimuli presented to different modalities). Results indicate that a plausible image (forward hand) *decreased* participants' ability to establish the temporal order of stimuli, as indicated by larger JNDs compared to an implausible image (inverted hand and arrow). These findings suggest that when a visual and tactile stimulus are more likely to have a common cause, this influences the degree of temporal integration.

Converging evidence for visuo-tactile integration depending on causal inference comes from Maselli, Kilteni, López-Moliner and Slater [27] who used a virtual reality set-up to investigate the impact of body contact and visual form on multisensory temporal integration specifically in the context of a body ownership illusion. Participants wore a head-mounted display which streamed a digital 3D replica of the room. By looking down, participants could see a gender-matched virtual body in the same location as their real body. Their first experiment compared a rotating wheel touching a virtual finger with a wheel that was separated from the virtual finger by 6 mm. Participants were asked to determine the temporal order of the visual stimulus (one full rotation of a virtual wheel) and the tactile stimulus (50 ms vibration to the participant's fingertip) presented at different SOAs. The results showed larger JNDs for the touching compared to the not-touching wheel condition. A second experiment tested if temporal order judgements are mediated by body-ownership by replacing the participants' virtual hands with virtual wooden sticks (manipulating body form). Results show that JNDs were larger in the context of a hand compared to a wooden stick. In addition, the virtual hand also increased the experience of ownership. It may be that the increased degree of temporal integration due to visual form facilitates ownership, or that the sense of ownership itself modulates the relative degree of integration, or both. Either way, together with the result by Ide and Hidaka [26], these findings suggests that a more plausible visual context for one's own body (e.g., indicated by hand form and orientation) increases the relative degree of multisensory temporal integration.

Conflicting findings, however, come from a study by Keys, Rich and Zopf [28]. This study also investigated the effect of viewed hand orientation on the degree of temporal integration but with a visuo-tactile asynchrony detection task. In the first experiment, participants viewed model hands in either anatomically plausible or implausible orientations. In a second experiment, the RHI was induced using synchronous touch in addition to the orientation manipulation to strengthen the multisensory cues indicating one's own body. In both experiments, participants detected short delays (40–280 ms) between a visual stimulus (a flash from an LED on the model hand) and a tactile stimulus (a tap to the fingertip of the participant's hidden hand). Each trial had two intervals, one in which the visual and tactile stimuli were synchronous and another where these were asynchronous. Participants indicated whether the asynchronous interval appeared first or second. The authors calculated and compared asynchrony detection thresholds between orientation conditions. They found that visual orientation cues did not influence detection accuracy for small visuo-tactile asynchronies, supported by Bayesian analyses to estimate the strength of evidence for the null result.

There are several potential explanations for the conflicting findings between these studies that investigated whether viewing visual body cues affects temporal integration. Keys et al. [28] used an asynchrony detection task whereas the two studies that reported an effect [26, 27] used a TOJ task. These tasks may pick up on different mechanisms [29]. There are also differences in terms of the ecological validity. Maselli et al. [27] used realistic visual and tactile virtual stimuli where an object touched the hand, whereas the other two studies had visual stimuli consisting of a light flash presented next to the viewed object (a 3D plaster hand or a drawing of a hand) combined with a tap on the participant's own hand. This could have affected whether the visual and tactile signals were perceived to 'belong together' and hence the degree of temporal integration. Finally, Keys et al. [28] used relatively large samples of 30 (Experiment 1) and 31 participants (Experiment 2) whereas both Ide and Hidaka [26] and Maselli et al. [27] used much smaller samples (12 –including the two authors—and 14, respectively) and it is unclear if these studies used a procedure to check for outliers. Current focus on methodological issues in experimental psychology and other sciences have emphasised the potential for false positive effects when using small sample sizes [30–33]. In sum, the discrepancy in findings regarding the modulation of visuo-tactile temporal integration of bodily signals is currently unclear; it could be due to task differences, variation in ecological validity and small sample sizes.

Findings also differ between studies investigating the effect of visual cues on the temporal integration of *visuo-tactile* stimuli reviewed above and those exploring *visuo-proprioceptive* bodily signals. Hoover and Harris [34] presented participants with videos of their own finger movements, either from a plausible or implausible perspective. By introducing small delays between the actual movement and visual feedback, they investigated whether visual body orientation cues modulate participants' temporal perception using an asynchrony detection task. Results suggested that viewing a hand from plausible perspective led to greater sensitivity for small temporal delays relative to an implausible viewpoint.

Similarly, a study by Zopf, Friedman and Williams [35] using a 3D virtual reality set-up found an effect of both visual form and orientation cues on visuo-proprioceptive temporal perception. These findings suggest that visual form and orientation cues indicating one's own body modulate temporal integration of visuo-proprioceptive inputs, and that these lead to smaller (instead of larger) thresholds to detect temporal delays. In other words, when multisensory inputs are perceived to belong to one's own body, temporal signals coming from visual and proprioceptive modalities seem to be *relatively more segregated*, resulting in a *better ability* to detect temporal delays. This different pattern of results raises the possibility that temporal processing of visuo-tactile and visuo-proprioceptive events may depend on different mechanisms (however, note that both studies investigating visuo-proprioceptive processing also used small samples of between 10 and 12 participants). It is therefore important to establish whether visual cues indeed modulate visuo-tactile temporal integration.

Visual cues seem to play an essential role in the spatial and temporal processing of bodily signals, but the exact mechanisms remain unknown due to limited and conflicting findings, especially in the visuo-tactile domain. Here, we investigate the processes underlying visuo-tactile integration of bodily signals, addressing some of the inconsistencies between previous studies. We tested if visual form and orientation cues modulate the degree of temporal integration with a visuo-tactile TOJ task, realistic visual and tactile stimuli, and pre-registered methods including a sampling plan based on Bayes factor (BF) thresholds for sufficiently large evidence [36, 37] and a procedure for outlier detection. Our research questions were: 1) Does the visual presentation of touch to a human hand compared to a non-body object modulate visuo-tactile TOJs? and 2) Does a plausible hand orientation compared to an implausible orientation modulate visuo-tactile TOJs? We hypothesised that if visual form and orientation

cues increase the degree of either temporal *integration* or *segregation*, then we would find larger or smaller JNDs respectively when viewing touch to a hand with a plausible orientation compared to a non-hand object or compared to an implausible orientation. In contrast, the null-hypothesis is that there is no effect of viewing condition on temporal integration as measured by JNDs.

## Methods

### Preregistration

We preregistered this study including the two hypotheses, planned methods and the data analysis plan before data collection. We followed this plan with two exceptions. First, we listed the subject pool as undergraduate students but also tested participants from the university community who responded to university advertising (e.g., postgraduate students). Second, for our Bayesian analyses we specified a predicted effect size based on previous studies including those that looked at the effect of visual cues on visuo-proprioceptive temporal integration. Based on comments from a reviewer, we subsequently changed the effect size calculation to only include studies that investigated visuo-tactile integration specifically. This does not change the predicted effect size (20 ms), so the actual results did not change due to this diversion. All materials, including the presentation script and videos, raw and summary data and analysis scripts, are available at (osf.io/grw57, registration date 06/06/2018).

### Participants

We planned to test 15 participants before calculating a BF to check if our data were sensitive enough to favour our alternative hypothesis over the null hypothesis or *vice versa* [37]. This initial sample size is based on previous studies that reported an effect of visual form or orientation cues on temporal integration [26, 27, 34, 35]. These studies all had sample sizes between 10 and 14 participants.

Conventionally, a BF (calculated as $BF_{10}$) larger than 3 indicates that the data provide moderate evidence for the alternative hypothesis and a BF smaller than 1/3 indicates that the data provide moderate evidence for the null hypothesis [38, 39]. A BF larger than 10:1 indicates strong evidence [39] and a BF smaller than 3:1 (i.e., BF between 1/3 and 3) indicates a lack of sensitivity to support one hypothesis over the other. We specified that we would continue data collection until results for both our research questions reached a BF larger than 10:1. If restricted by time, we planned to accept a BF larger than 3:1 as sufficient evidence. We continued data collection after the first 15 participants as the result for one of the two research questions had not yet reached a BF larger than 10:1. Then, due to time restrictions, we finished testing before the results converged on a BF larger than 10:1; they did reach the second level of 3:1.

Our sample before exclusions consisted of 42 right-handed individuals with normal or corrected-to-normal vision (mean age = 26.5 years, age range = 18–58, SD = 9.1 years, 27 female). Our final sample after exclusions is reported in the results section. Participants provided written consent and received $15 per hour for participation. The Macquarie University Human Research Ethics committee approved the study.

### Stimuli

**Visual stimulus.** For the visual stimuli, we manipulated the object that was being touched. These consisted of videos depicting touch to (a) a human hand in a plausible orientation; (b) a sponge; and (c) a human hand in an implausible orientation (see Fig 1A). To avoid

# A Visual Stimuli

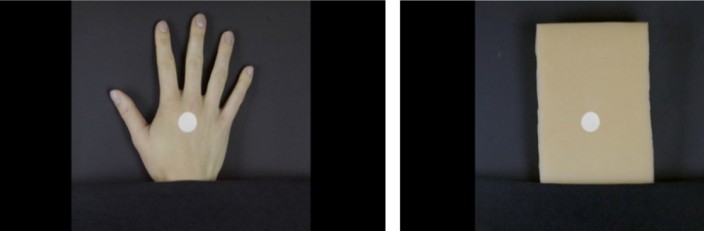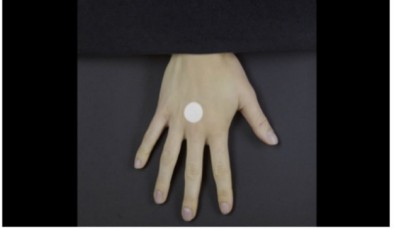

# B Experimental Setup

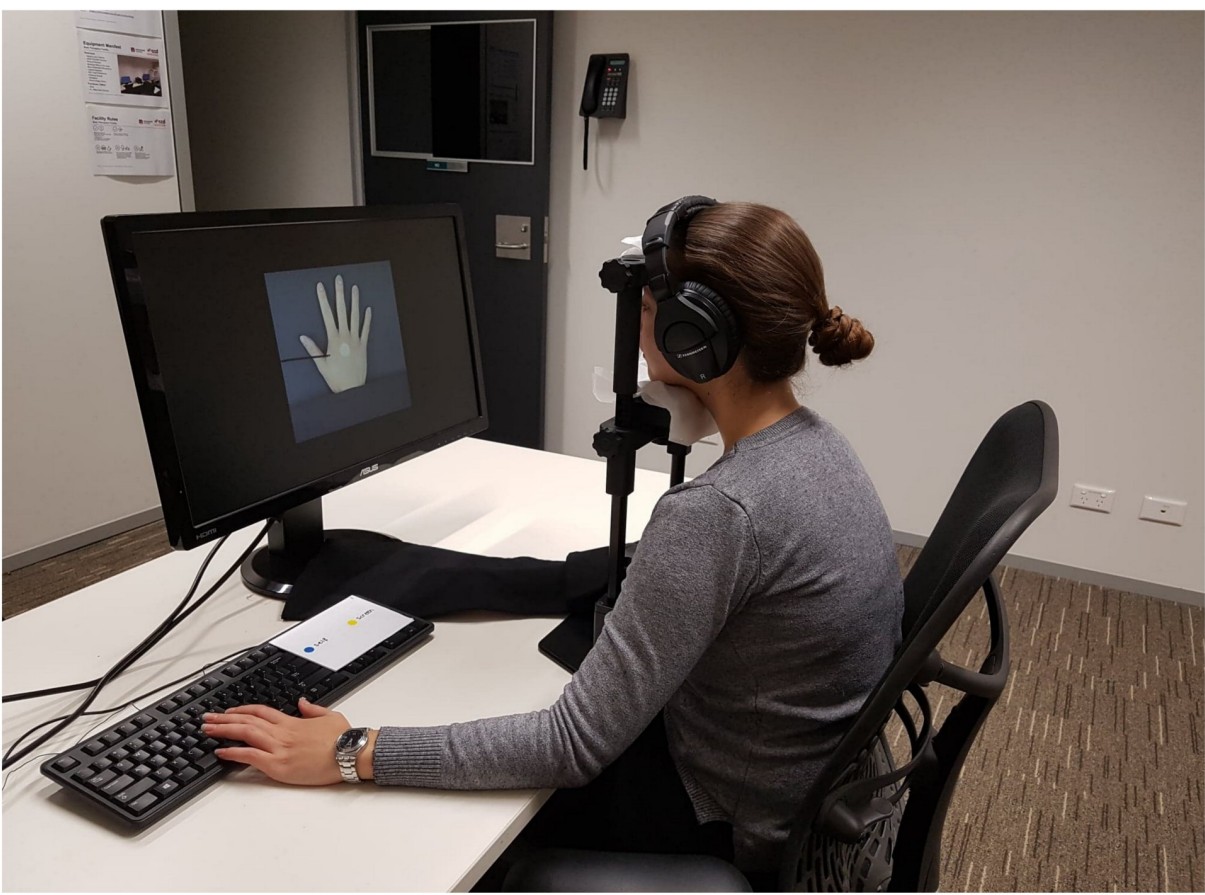

**Fig 1. Visual stimuli and experimental setup.** (A) Screenshots of the videos depicting touch to either a human hand in a plausible orientation (left), a sponge (middle) or a human hand in an implausible orientation (right). Examples of each of the three condition videos can be found on the OSF page for this project: osf.io/grw57. (B) Participants watched videos of a hand or sponge being touched and indicated whether the seen touch or the felt touch came first by pressing one of two response keys with their left hand. To mask any noise from the tactile stimulator, participants listened to white noise via headphones. Viewing distance was kept constant with a chin rest.

confounding factors, we used DaVinci Resolve video editing software to create videos that were identical in terms of the timing, movement and location of the touch.

First, we filmed two videos, one of a stationary human hand and one of a stationary sponge. We used actual video footage over a static image was to increase the visual realism, for

example, due to very small jitter and blood flow in the hand. We used a sponge for the control condition as it depresses to touch in the same way as a hand. The sponge was matched with the hand (size, light, shade, colour and white balance) so that it differed only in form.

Second, to create a realistic touch event where the surface was slightly indented upon touch, we filmed a black stick touching a green patch (16 mm in diameter) placed on a soft material. This meant that after editing it actually looked as if the hand and the sponge were being touched but the touch was identical across conditions. Another benefit of showing an indentation is that the exact moment of touch stands out more prominently, which is important for the TOJ task. This recreation of real touch could only be achieved by filming the stick touching a coloured surface so that it could later on be etched and separated from the background with the shadows intact. We changed the colour of the green patch to be white in the editing software, so there was less visual difference.

Third, we overlaid a hand coming into the screen and touching a white dot with a stick onto the video of the stationary hand and sponge. We created the implausible hand orientation video by flipping the video with the plausible hand orientation and then overlaying the touch video.

Finally, we cut each video into separate frames using MATLAB (MathWorks, Natick, Massachusetts, U.S.A) so we could control the onset and presentation of the video frames relative to the tactile stimulus across the three conditions. All presented videos consisted of 62 frames (presented for 33.33 ms per frame) resulting in approximately two-second videos. To keep participants focused and reduce expectancy effects, we varied the exact moment at which the visual touch happened in each video. Each trial consisted of 62 frames but we randomly moved the start and end frame by 0–5 frames forwards or backwards in the sequence. The five frames on either side of these 62 frames depicted a stationary hand or sponge (before and after the stick has come in for the touch) so for each trial a video showed a complete touch action. These were presented on an ASUS monitor which was 60 cm wide and 34 cm high with a refresh rate of 60 hz. The hands and sponge presented on the screen were 19 cm high and were presented at a viewing distance of 48 cm, resulting in a visual angle of 22 degrees. The vertical distance between the top of the participant's hand and the bottom of the hands and sponge on the screen was approximately 15 cm.

**Tactile stimulus.** The tactile stimulus was a 30 ms pulse of the vibrotactile stimulator, which applied a tap to the back of the participant's hand via an electromagnetic solenoid-type vibrotactile stimulator (diameter: 18 mm and probe height: 12 mm, Dancer Design, St. Helens, UK; dancerdesign.co.uk) and an amplifier (TactAmp 4.2 with a D25 serial port, Dancer Design).

Before the start of the experiment, the tactile stimulator was attached to the back of the participant's right hand, just below the junction between the hand and the middle finger (the metacarpophalangeal joint), and covered with black fabric to hide it from view. A printed screenshot of the hand video was initially placed next to the participant's right hand to match the relative location of the tactor as closely as possible to the relative location of observed touch on the screen. The touch stimulus closely matched the visual touch from the stick in the videos in terms of phenomenology and duration.

To mask any noise from the tactor, participants listened to white noise via around the ear, closed-back headphones (Sennheiser HD 280 pro, 64 ohm). We presented and controlled stimulus presentation with MATLAB and the Psychophysics Toolbox [40, 41].

## Task and procedure

Participants removed jewellery from their hands, and then sat in front of a computer screen with their chins on a chinrest. The participant's right hand was placed in front on the table and

aligned with the middle of the computer screen (see Fig 1B for the experimental setup). Participants responded by pressing the H or J key on a keyboard with the middle and index finger of the left hand. A yellow and blue sticker (corresponding to either 'visual-first' or 'tactile-first') were placed over the H and J response keys. To eliminate the effect of any response bias, half the participants responded to the 'visual-first' cue with their index finger and to the 'tactile-first' cue with their middle finger, and for the other half of participants this was the other way around. A description of the responses was placed next to the response keys as a reminder for the participants.

During each trial, participants watched one of the three videos and felt a touch on their own right hand. The tactile stimuli were presented with one of the following SOAs relative to the visual touch: ± 33, ± 67, ± 100, ± 133, ± 167 ± 200 ± 333 ms (negative values represent a tactile-leading stimulus and positive values show a visual-leading stimulus). The task was to determine for each trial whether the visual touch on the screen or the felt touch on one's hand came first (forced choice). Inter-trial timing was jittered to reduce the likelihood of participants getting into a routine response. The next trial started 800~1200 ms after a response was recorded. There were scheduled breaks and a participant could choose to take a break or continue the experiment by pressing the space bar.

Participants performed a practice block to become familiar with the task and the button responses. On each of the practice trials participants received feedback on whether a response was correct or incorrect. For the practice block, the participants completed 84 trials with each SOA presented twice per condition (randomly intermingled). During the experiment, participants did not receive feedback on their performance. For each of the three conditions, there were 20 trials with 14 different SOAs resulting in 840 trials in total presented in a random order.

The practice run and experiment combined took approximately one hour to complete. After the experiment, participants filled out three questionnaires which took another 20 minutes (see additional measures section below).

## Data analysis

For our main analysis, we compared the mean JNDs between the experimental conditions across the group. To calculate each participant's individual JNDs, we first calculated the proportion of 'visual-first' responses for each SOA in each condition. For example, 0 = 0% 'visual-first'/100% 'tactile-first' responses and 100 = 100% 'visual-first'/0% 'tactile-first' responses. We then fitted a sigmoid function (cumulative Gaussian distributions) to the proportion of 'visual-first' responses using maximum likelihood estimation using the MATLAB Palamedes toolbox (Prins and Kingdom [42]; www.palamedestoolbox.org). We then calculated each participant's JND for each of the three conditions from the fitted psychometric functions, using the formula: JND = (75% point—25% point)/2.

Another measure often reported is the participants' point of subjective simultaneity (PSS), which is the SOA at the 50% crossover point (the same number of 'visual-first' and 'tactile-first' responses) [43]. Here, the PSS does not provide relevant information about the strength of coupling between the visual and tactile stimuli [23], so we have not included it, but we present these data on the OSF page for this study as it may be useful for meta-analyses.

We excluded data sets for participants based on three criteria: (1) a JND larger than three standard deviations from the group mean in any condition; (2) any of the three curves (representing the three conditions) failing to converge on a solution for fitting a sigmoid function (in the excluded subjects there are clear straight lines instead of the expected convergence on

sigmoid functions); or (3) incomplete data due to a technical error or failure to perform the task.

We performed Bayesian t-tests, which require a specification of the theory which is tested against the null hypothesis (i.e., the probability of different effects given the alternative theory [37]). We specified a plausible predicted effect size based on the findings by Ide and Hidaka [26] and Maselli et al. [27] which resulted in an average of 20 ms. We set our predicted effect size at 20 ms in either direction. We used the same informative prior for both our hypotheses and specified a normal distribution with the mean set to zero, the standard deviation set to 20 and the tails set to two-sided. We calculated the BF in MATLAB with a script downloaded from Dienes' online calculator (http://www.lifesci.sussex.ac.uk/home/Zoltan_Dienes/inference/Bayes.htm). An informative prior (as opposed to a non-informative or default prior) contains information that is relevant to the estimation of the model and consequently the choice of prior can impact the final estimates [44]. To check to what extent our final estimates depend on our informative prior, we also performed a robustness check with the statistical software JASP (www.jasp-stats.org) using JASP's range of default priors. We used a Cauchy prior for the size of the effect and specified a non-directional alternative hypothesis.

We planned to sequentially plot the BFs as data were collected, which does not require any corrections for multiple comparisons [45]. This produces a visual impression of when the BFs are converging on a specific level of evidence which favours one hypothesis over another [46]. To visualise the data, we used R including the package ggplot2 [47].

## Additional measures and exploratory analyses

Mirror-touch synaesthesia (MTS) is a condition where observing touch to another person results in an experience of touch on the observer's own body [48]. It is possible that multisensory interactions are generally enhanced in synaesthesia (including MTS) which could potentially result in an increased degree of temporal integration [49, 50]. To ensure we were able to detect and exclude MTS participants, we used short synaesthesia questionnaires. The first questionnaire screened for synaesthesia in general, the second for MTS specifically (adapted from Ward, Schnakenberg and Banissy [51]). Our adapted MTS screener consisted of 20 of the Ward et al. [51] short video clips depicting touch to a human or an inanimate object and four videos of someone scratching their chest or upper arm. After each video participants were asked whether they experienced anything on their body, how they would describe the sensation and where on their body it was felt. We planned to analyse the data both with and without potential mirror-touch synaesthetes to see if this influenced the outcome.

Finally, we planned to explore if, in our sample, larger JNDs for the hand condition compared to the sponge condition correlated with higher empathy scores. Ward et al. [51] tested for different categories of empathy: cognitive empathy (predicting others' thoughts and feelings), social skills (being able to interact with others appropriately) and emotional reactivity (intuitively understanding how people feel). They found that the latter was increased in MTS. This type of empathy could also be related to increased temporal integration in non-synaesthetes. To explore this, we had participants fill out a short empathy questionnaire [52], which was divided into the same three categories as previously used by Ward et al. [51].

## Results

### Exclusions

Data from six participants were removed due to a JND larger than three standard deviations from the group mean for at least one of the three conditions. For four out of the same six participants (but not for any of the others) the condition fits also failed to converge on a solution

for a sigmoid function. Two participants had incomplete datasets due to a technical error and one participant failed to perform the task. After the exclusion of these nine participants, there were 33 participants (mean age = 26.3 years, age range = 18–55, SD = 8.3 years, 20 female) included in the analysis.

## Just-noticeable difference (JND)

We calculated the average proportion of 'visual-first' responses (derived before fitting the data) and plotted this against SOAs for the three conditions. As can be seen in Fig 2A, the averaged proportions were similar across the three conditions. Fig 2B depicts both participants' individual JNDs (derived from individual data fits) and the group mean JNDs for the three conditions. This shows that the mean JNDs are very similar for the three conditions: hand plausible orientation mean JND = 104 ms, SD = 45 ms, 95% CI [88, 119], hand implausible orientation mean JND = 102 ms, SD = 42 ms, 95% CI [87, 116] and sponge mean JND = 102 ms, SD = 46 ms, 95% CI [86, 118]. Fig 2C depicts the condition differences. The mean difference for the plausibly orientated hand versus sponge comparison was 1.39 ms, 95% CI [-3.50, 6.28] and for the plausibly versus implausibly orientated hand comparison this was 1.48 ms, 95% CI [-4.94, 7.88]. Fig 2C also shows that there was considerable variability for the JND differences between participants, with differences ranging from -40 to 29 ms for the form and -29 to 32 ms for the orientation comparison.

## Bayesian analysis

We calculated the $BF_{10}$ with a Bayesian t-test to test if visual cues in terms of form and orientation modulate visuo-tactile temporal integration. First, to test for the effect of form, we compared the group mean JND between the plausible hand orientation and sponge condition. Fig 3 depicts the development of the BF as a function of the number of participants tested. As data accumulates with more participants being tested, evidence converges and the BF increasingly supports the null hypothesis indicated by values smaller than 1 [53]. Results indicate moderate evidence (BF = 0.178) that body form did not have an influence on the precision of TOJs. In other words, the data were 5.61 (1/0.178) times more likely under the null hypothesis than the alternative hypothesis. It is interesting to note that the BF in Fig 3 (plausible hand orientation versus sponge) does cross the BF > 10 threshold with less than twenty subjects. However, this is likely due to noise as data points have not converged, potentially because of the variability in the data [46]. This emphasises the importance of having larger sample sizes. We next tested if hand orientation modulates visuo-tactile TOJs by comparing the group mean JND between the plausible hand orientation and implausible hand orientation (see Fig 3). A Bayesian t-test indicates moderate evidence (BF = 0.144) that the JND for the two conditions is not different. In other words, the data were 6.94 (1/0.144) times more likely under the null hypothesis than the alternative hypothesis. To check the impact of the prior on our analysis, we also performed an analysis and robustness check in JASP using the default priors. This indicated that evidence for the null hypothesis is stable across a range of specified parameters which suggests that our analysis is robust.

## Questionnaires (planned exploratory research)

To assess MTS, participants filled out a screening questionnaire. Two participants had a score >7 (they scored 10 and 12), which Ward et al. [51] suggest is a potential indication of MTS. To ensure that the data from these two participants did not influence our outcome, we also calculated the BFs without data from these two participants, which did not change the pattern of

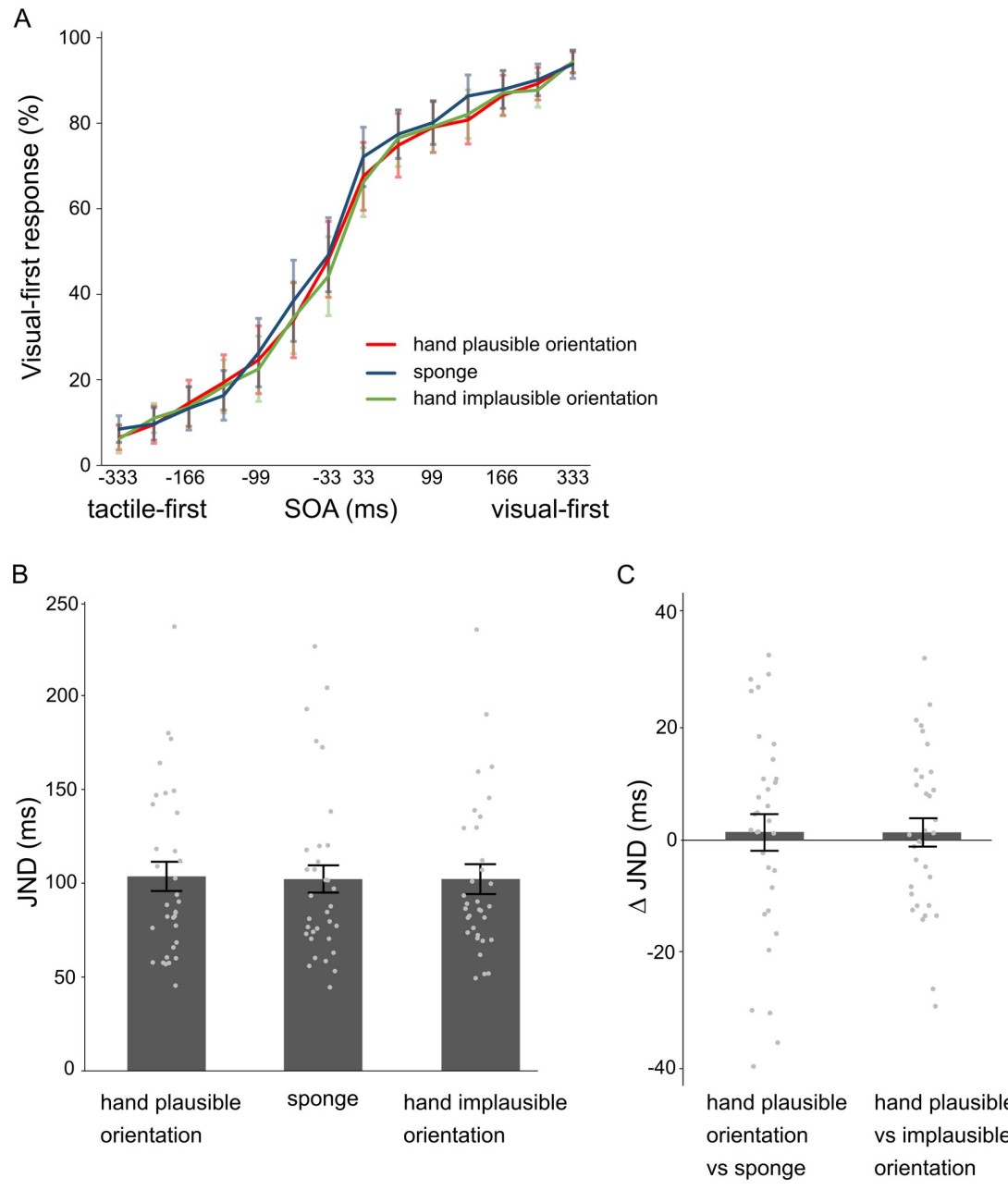

**Fig 2. 'Visual-first' responses and JND results.** (A) Averaged proportion of 'visual-first' responses plotted against SOAs for the three conditions. The averaged 'visual-first' responses for the different conditions are very similar. Note that we fitted sigmoid functions to the data of each participant and then calculated JNDs for each participant and condition. (B) Bar graph showing mean JNDs for each of the three conditions with JND estimates from individual participants indicated by scatter points. (C) Bar graphs showing the mean JND differences for the two comparisons. JND differences from individual participants are indicated by scatter points and show high between-subject variability (ΔJND form = JND hand plausible orientation–JND sponge and ΔJND orientation = JND hand plausible orientation–JND hand implausible orientation). Error bars represent 95% CI.

results (mean difference plausible hand orientation vs. sponge = 1.53 ms, BF = 0.19, mean difference plausible hand orientation vs. implausible hand orientation = 1.89 ms, BF = 0.17). The short synaesthesia questionnaire did not pick up other types of synaesthesia in any of the participants.

hand plausible orientation
versus sponge

hand plausible orientation
versus implausible orientation

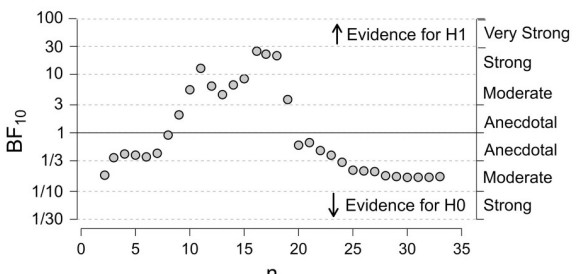 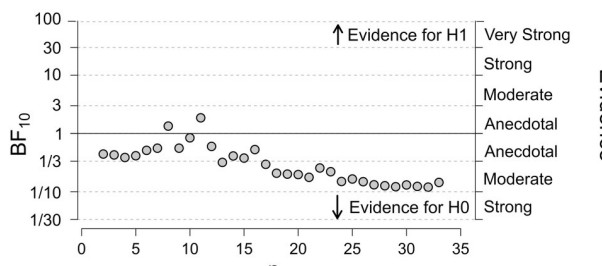

**Fig 3. Sequential plotting of the Bayes factors.** We compared the two conditions: plausible hand orientation versus sponge (left panel) and plausible hand orientation versus implausible hand orientation (right panel). Sequential plotting of the BF shows that the BFs start to converge at N = 20 (plausible hand orientation versus sponge) and N = 24 (plausible hand orientation versus implausible hand orientation), which in both cases provides moderate evidence to support the null hypothesis.

We also explored if larger JNDs for the hand condition compared to the sponge condition correlated with higher empathy scores. A Bayesian analysis demonstrated positive correlations for cognitive empathy (r = 0.36, BF = 1.69) and social skills (r = 0.39, BF = 2.47) but not for emotional reactivity (r = -0.14, B = 0.29). However, the BFs indicate a lack of sensitivity and more research is needed to further investigate this.

## Discussion

To successfully interact with the environment, the brain needs to establish which unisensory signals belong to the same object or event despite small intersensory spatiotemporal differences that may be present [11, 12]. A Bayesian causal inference model proposes that multisensory perception relies on causal inference to establish the probabilities as to which unimodal inputs share a common source and should therefore be integrated (or otherwise segregated due to separate sources). This framework might also apply to the processing of spatial and temporal signals from visual, tactile and proprioceptive modalities to establish body ownership [24]. However, limited and conflicting empirical evidence exists regarding the temporal integration of bodily signals.

In this study we used a TOJ task to test whether viewing touch to a hand with a plausible orientation compared to a non-hand object and compared to a hand with an implausible orientation modulates the relative degree of visuo-tactile temporal integration. We hypothesised that if visual form and orientation cues influence the degree of either temporal *integration* or *segregation*, this would result in larger or smaller JNDs respectively when comparing the three conditions. Results show that group mean JNDs were the same across our three conditions. Bayesian analysis further indicates moderate evidence that neither body form (BF = 0.178) or orientation (BF = 0.144) had an influence on the precision of TOJs. These findings suggest that plausible visual cues do not modulate the degree of integration (or segregation) of visual and tactile bodily inputs.

Previous studies that investigated the effect of visual cues on the degree of visuo-tactile temporal integration have reported conflicting findings [26–28]. Keys et al. [28] used an asynchrony detection task, and found no effects. In contrast, two studies using a TOJ task reported an effect [26, 27]. Asynchrony detection involves simply detecting a temporal delay (irrespective of order) whereas TOJ tasks involve detecting the specific temporal order of stimuli. It is

possible that only tasks involving temporal order are affected by the plausibility of the context for the multisensory stimuli. Here, we used a TOJ task to test this possibility, but also found evidence for no effect. This suggests that the reported results do not depend on the type of task.

As Bayesian causal inference models propose that the degree of multisensory integration is influenced by a causal inference process that determines the relative probability that signals belong together or not [19–21], a realistic pairing between a visual and tactile touch stimulus might be important to observe modulations in multisensory integration. A study might therefore fail to show an effect if the stimuli are normally unlikely to co-occur, such as in the Keys et al. [28] study. In that study, the visual stimulus was a light presented next to a 3D plaster hand, paired with a tactile stimulus on the participant's own hand. Outside of the laboratory, this is not a common co-occurrence (light flashes next to our hands together with a touch), and so these stimuli may not have been integrated as 'belonging together'. In contrast, Maselli et al. [27] used realistic stimuli, where participants observed a touch to a virtual hand and felt a presumably similar touch. However, Ide and Hidaka [26] reported an effect with artificial stimuli, consisting of a light presented on a line drawing of a hand paired with a tactile touch, suggesting that these stimuli should be sufficiently realistic to modulate temporal integration. An important motivation for the current study was to test the effect of visual cues on temporal integration with a realistic pairing between the visual and tactile stimulus. Our videos were of a real touch to a human hand along with a tactile tap that matched in terms of size, shape, pressure and duration. Thus, our results suggest that even when the visual and tactile stimulus more realistically 'belong together' (compared to Keys et al. [28] and Ide and Hidaka [26]), causal inference does not modulate the degree of temporal integration.

Another aspect that could have influenced the ecological validity is whether the visual hand stimulus is presented in 2D or 3D. Both Ide and Hidaka [26] and Maselli et al. [27] reported an effect with 2D and 3D stimuli respectively whereas the current study and Keys et al. [28] found evidence for the null with 2D and 3D hands. Further, previous studies show that simple 2D hand images are sufficient to establish interactions between visual and tactile stimuli [26, 54, 55]. Thus, it does not seem to be the dimensionality of the visual hand that determines the effect (or lack thereof). In addition, we presented the hand on a computer screen which results in a mismatch in the orthogonal rotation between the presented hand and the participant's own hand. This could have influenced our results, however this seems improbable as Keys et al. [28] presented their hand stimuli in the same orthogonal rotation as the participant's hand and also reported evidence for the null. Future studies could investigate any potential effect of the orthogonal rotation of the hand on temporal integration.

Finally, it is possible that causal inference only affects the degree of temporal integration when the visual and tactile stimuli come in physical contact with each other. Questionnaire results in the study by Maselli et al. [27] showed that participants felt as if their index finger was actually being touched by the virtual wheel, suggesting an explicit inference of a common cause. This might be the reason why this study found causal inference effects on temporal integration. In sum, the discrepancy in findings is unlikely due to variation in the ecological validity of the stimuli. However, it is possible that visual cues only modulate visuo-tactile TOJs when the visual and tactile stimulus join in space and this could be further tested in future experiments.

The conflicting findings may also be influenced by the relative sample sizes. Both the current study and the Keys et al. [28] study used relatively large sample sizes combined with a clearly defined sampling plan to ensure sensitive data for hypothesis evaluation and a procedure to check for outliers. Studies with high power and a complete analysis strategy are typically more robust and therefore less easily influenced by outliers and analytical flexibility [30–

32]. In addition, our results show large individual differences between JNDs, ranging from -40 to 29 ms for the form comparison and -29 to 32 ms for the orientation comparison (see Fig 2C) and this was also the case in the study by Maselli et al. [27] (range of JNDs between ~-50 to ~100 ms for Experiment 1 and ~-50 to ~50 ms for Experiment 2, see their Figs 2B and 3A; Ide and Hidaka [26] did not report on individual participant data). Noise due to data variability may strongly affect outcomes on a TOJ task, especially when sample sizes are small [30, 31, 56]. To make sure our experiment produced sensitive data to either support our alternative or null hypothesis we based our sample size on a Bayesian analysis. Of course, it is always possible that there is an effect that is so small in the context of the variance within the task and participants, that we have not detected it. However, our methods give us considerable evidence for there being no effect of visual cues on the degree of visuo-tactile temporal integration of bodily signals.

Our findings suggest a difference between the temporal processing of visuo-*tactile* and visuo-*proprioceptive* bodily stimuli. Hoover and Harris [34] and Zopf et al. [35] demonstrated that visual form and orientation cues led to greater sensitivity to detect small temporal delays, which suggests that visual plausibility leads to relatively more segregation of visual and proprioceptive temporal inputs. Our data, however, together with Keys et al. [28], show that such visual cues do not modulate visuo-tactile integration. Thus, it appears that different mechanisms could underlie the temporal processing of visuo-tactile and visuo-proprioceptive bodily inputs.

A Bayesian causal inference model for body ownership suggests that evidence for a common source influences the degree of temporal and spatial integration of visual, tactile and proprioceptive inputs [24]. Findings in the temporal domain that show no effect on the degree of temporal integration (the current study and [28]) and those showing increased temporal segregation [34, 35], do not fit within this general framework. On the other hand, studies with the RHI do suggest that plausible visual cues can promote spatial integration of visuo-proprioceptive bodily signals. In addition, findings from crossmodal congruency tasks also show effects of visual form and orientation cues on visuo-tactile spatial interactions which might also indicate spatial integration [54, 55, 57, 58] (although this is debated as effects could merely occur on a response level [59, 60]). Evidence for a Bayesian causal inference framework for the integration of bodily signals therefore comes predominantly from the spatial domain (although see more recent findings by Motyka and Litwin [61]). Table 1 provides an overview of selected papers that investigated the effect of visual cues on temporal and spatial integration or segregation of bodily signals.

The measured JNDs in the current study are relatively large compared to previous studies that used more basic stimuli (e.g., Keetels and Vroomen [62]). This could be due to the complex nature of the visual stimuli as it might be harder to extract their onset compared to simple stimuli such as light flashes. Sensitivity for temporal order on a TOJ task has generally been shown to deteriorate when more complex stimuli are used [27, 63]. This sensory temporal noise in the visual touch signal could influence the probability estimate for a common cause for the visual and tactile touch signals [15]. It is thus possible that any additional influence from the visual cues (form and orientation) on the the assumption of a common cause (i.e., common cause prior) might have been too small and not measurable in the context of the stimuli we used. In other words, a Bayesian causal inference mechanism for temporal integration which takes into account body form and orientation could in principle still hold up and further testing and computational modeling would be required to evaluate the evidence for this model. However, an important motivation for our study was to use naturalistic stimuli to investigate temporal processes for body perception and our results provide evidence that

**Table 1. Selected papers that directly test visual context (form and orientation) effects on temporal and spatial integration of visual, tactile and proprioceptive bodily inputs.**

| Studies | Form[a] | Orientation | Method[b] | N/expt. | Temporal effects | | Spatial effects | |
|---|---|---|---|---|---|---|---|---|
| | | | | | Visuo-tactile | Visuo-proprioceptive | Visuo-tactile[c] | Visuo-proprioceptive |
| Current study | ✓ | ✓ | TOJ | 33 | No integration or segregation | | | |
| Keys et al. [28] | | ✓ | SJ | 30, 31 | No integration or segregation | | | |
| Ide and Hidaka [26] | ✓ | ✓ | TOJ | 12 | Integration | | | |
| Maselli et al. [27] | ✓ | | TOJ | 14 | Integration | | | |
| Zopf et al. [35] | ✓ | ✓ | SJ | 10, 12, 11 | | Segregation | | |
| Hoover and Harris [34] | | ✓ | SJ | 10 | | Segregation | | |
| Igarashi et al. [58] | ✓ | | CCT | 22 | | | Integration | |
| Igarashi et al. [55] | | ✓ | CCT | 8, 12 | | | Integration | |
| Pavani et al. [57] | ✓ | ✓ | CCT | 10, 10 | | | Integration | |
| Tsakiris et al. [8] | ✓ | | JHP (RHI) | 40 | | | | Integration |
| Holmes et al. [25] | ✓ | ✓ | Reaching | 18, 21, 24, 12, 12 | | | | Integration |
| Tsakiris and Haggard [6] | ✓ | ✓ | JHP (RHI) | 8, 8, 10,14 | | | | Integration |
| Costantini and Haggard [9] | | ✓ | JHP (RHI) | 16 | | | | Integration |

[a] Significant (p < 0.05 or equivalent) report of visual form and/or orientation cues influencing temporal and spatial integration/segregation.

[b] Temporal order judgement (TOJ) task, synchrony judgement (SJ) task, crossmodal congruency task (CCT), judgement of hand position (JHP), rubber hand illusion (RHI).

[c] The research into visuo-tactile spatial interactions, referenced here, involves crossmodal congruency tasks, and it is not clear whether these results also indicate spatial integration or merely an effect at the response selection level.

under these conditions, the relative degree of temporal integration is not modulated by the body form and orientation context.

Shimada et al. [5] also used naturalistic body stimuli and showed that participants still integrate bodily signals and experience a strong bodily illusion even for relatively large temporal differences between viewed and felt touch (300 ms). In contrast, here we show that naturalistic touch stimuli can on average be reliably separated when there is a 104 ms difference. This suggests that even when signals are processed as being temporally separate, in bodily illusions these might still be perceptually integrated on a non-temporal dimension. Perception in bodily illusions is likely influenced not just by temporal cues but also by other types of information such as the plausibility of the visual context and factors such as temporal recalibration or attention [6, 43]. This may result in the observation of less strict temporal constraints as reported by Shimada et al. [5]. Our finding suggests that the visual context on its own does not modulate the degree of temporal integration.

Conflicting findings regarding an effect of causal inference on the degree of temporal integration have also been reported in the audio-visual domain. For example, in the case of speech perception, Vatakis and Spence [64] showed with a TOJ task that temporal lags between auditory and visual stimuli are less likely to be perceived when there is other evidence that the stimuli belong to the same multisensory event. Participants were presented with videos and sounds at different SOAs that were either congruent or incongruent in terms of gender (e.g., a female face combined with either a female or male voice, respectively) and phonemic content

(e.g., a female pronouncing /ba/ combined with the sound /ba/ or /da/). JNDs were larger for both congruent gender and phonemic content compared to incongruent content, which indicates less precision. These results suggest that a plausible context that is consistent with a common cause for auditory and visual speech stimuli can modulate the degree of temporal integration. Further, Parise and Spence [23] demonstrated that certain correspondences between properties in different modalities can increase the degree of spatial and temporal integration of auditory and visual stimuli not related to human speech. In a TOJ task, the authors presented participants with pairs of spatially or temporally discrepant auditory and visual stimuli that were either matched (e.g., a small circle presented with a high-pitched tone) or mismatched (e.g., a small circle presented with a low-pitched tone). Results showed larger JNDs for matched compared to mismatched stimuli, suggesting reduced access to intersensory conflicts. The authors conclude that crossmodal correspondences between the stimuli increased spatial and temporal integration of auditory and visual non-speech signals.

Other studies however, have not been able to find a causal inference effect on the degree of temporal integration of auditory and visual non-speech stimuli. For instance, Vatakis, Ghazanfar and Spence [65] used a TOJ task and presented participants with matched and mismatched audio-visual stimuli which consisted of: 1) one call-type made by two different monkeys, 2) two different call-types from the same monkey, 3) a person versus monkey 'cooing,' and 4) speech sounds produced by a male and a female person. Results show that participants' performance was only affected in the case of human speech stimuli, not when observing matched versus mismatched monkey calls or human imitations of monkey calls (see also Vatakis and Spence [66] for similar findings). This suggests that contextual information could modulate audio-visual temporal integration, but that this might be limited to certain stimuli. To better understand multisensory processing, more research is needed across different modalities to clarify how certain properties between multisensory events may or may not affect temporal integration.

Thus overall, the influence of high-level contextual cues on the degree of temporal integration remains unclear. However, the major factors that are thought to modulate how multisensory events are integrated in time are the temporal and spatial proximity of signals [43, 67]. Studies show for example that increased spatial separation between cross-modal signals improves sensitivity for temporal order in TOJ tasks [68–70]. One explanation is that closeness in space indicates a single underlying cause, which increases relative intersensory binding and limits access to the temporal properties of individual signals. Alternatively, when there is a spatial gap between stimuli, participant could also rely on spatial position as an additional cue to establish which modality was presented first. This in turn could lead to smaller JNDs compared to when there are no or very small spatial discrepancies [68]. With respect to our results, the spatial distance between the viewed object and the hidden hand could be another potential cause for the discrepant findings (also see Keys et al. [28] for a similar discussion). It is important to note, however, that our spatial differences are less than half the distance of studies that found positive effects (Ide and Hidaka [26], 30–50 cm). Future research could systematically investigate the role of spatial distance on temporal effects and how these spatial cues might interact with contextual body cues.

## Conclusion

Our study provides compelling evidence, supported by Bayesian analysis, that visual form and orientation cues do not modulate the degree of *temporal integration* of *visuo-tactile* bodily signals. This finding is in contrast to findings for *visuo-proprioceptive* bodily signals, in which visual plausibility results in enhanced *segregation*. Thus, it appears that the underlying

temporal mechanisms are different for visuo-tactile and visuo-proprioceptive integration. Bayesian causal inference models propose that evidence for a common source increases the degree of both spatial and temporal integration of multisensory inputs. Studies with the RHI indeed show that a plausible context can promote the inference of a common cause (one's body) resulting in *spatial integration* of *visuo-proprioceptive* signals. Conversely, the current and previous findings suggest that a plausible context does not result in an increased degree of temporal integration of bodily signals and hence are inconsistent with the idea that cues indicating oneself modulate the integration of multisensory signals in the temporal domain. Further, research into the effects of higher-level cues such as context on temporal integration is limited and conflicting in the multisensory literature more broadly. Thus generally, more research, including computational modelling, is needed to provide better insight into the factors that influence how multisensory events are integrated.

## Author Contributions

**Conceptualization:** Sophie Smit, Anina N. Rich, Regine Zopf.

**Data curation:** Sophie Smit.

**Formal analysis:** Sophie Smit, Regine Zopf.

**Writing – original draft:** Sophie Smit.

**Writing – review & editing:** Sophie Smit, Anina N. Rich, Regine Zopf.

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
