## [Decision Letter · Decision Letter 0]

9 Jul 2019

PONE-D-19-14549

Visual body form and orientation cues do not modulate visuo-tactile temporal integration

PLOS ONE

Dear Dr Smit,

Thank you for submitting your manuscript to PLOS ONE. I have now obtained two reviews of your paper. As you will see, both reviewers are positive about your study and believe that it should be published, as do I. Both reviewers have made a number of comments and suggestions about how your study could be improved. I would like to invite to you revise your paper in light of these comments, which I do not think should be difficult.

We would appreciate receiving your revised manuscript by Aug 23 2019 11:59PM. To enhance the reproducibility of your results, we recommend that if applicable you deposit your laboratory protocols in protocols.io, where a protocol can be assigned its own identifier (DOI) such that it can be cited independently in the future. For instructions see: http://journals.plos.org/plosone/s/submission-guidelines#loc-laboratory-protocols

We look forward to receiving your revised manuscript.

Kind regards,

Matthew Longo, Ph.D.

Academic Editor

PLOS ONE

Journal Requirements:

1.

Reviewers' comments:

Reviewer's Responses to Questions

**Comments to the Author**

1. Is the manuscript technically sound, and do the data support the conclusions?

Reviewer #1: Partly

Reviewer #2: Yes

2. Has the statistical analysis been performed appropriately and rigorously? 

Reviewer #1: I Don't Know

Reviewer #2: Yes

3. Have the authors made all data underlying the findings in their manuscript fully available?

Reviewer #1: Yes

Reviewer #2: No

4. Is the manuscript presented in an intelligible fashion and written in standard English?

Reviewer #1: Yes

Reviewer #2: Yes

5. Review Comments to the Author

Reviewer #1: Summary of the Research and Overall Impression

- The focus of this research is centered around the question of how sensory cues modulate multisensory integration for body perception. In particular, this study used a visuo-tactile temporal order judgment (TOJ) task to test whether visual form and orientations cues could cause temporal integration. Comparisons were made between three conditions (hand in a plausible orientation; hand in an implausible orientation; sponge as a non-hand object)

- The authors provide an extensive literature review that looked at various aspects of multisensory integration. Individual studies are described in quite a lot of detail but are used to relate and put the current study into context with previous findings.

- Potential issues and limitations of previous studies were effectively identified and they aimed to remove some of these in the current study.

- The findings showed that the three conditions did not differ, implying that visual form and orientation cues do not modulate temporal integration of visuo-tactile bodily signals. The authors focus the discussion on the conflicting findings in previous studies and conclude that the underlying mechanisms involved in multisensory integration appear to be different depending on the modalities used.

Examples and Evidence

Major Issues:

- 1) Since the introduction jumps right into a detailed review of the literature, it was a bit unclear what the current study was going to look at until the final paragraph. This made it harder to follow the logic of how and why the research questions fit into existing literature.

- 2) The discussion repeats a lot of what was said in the introduction about the limitations and how the studies differ. A lot of those details included in the introduction about each study could be moved to the discussion to provide more support for the claims and conclusions made. This would also help to streamline the introduction and make it easier to follow what the current study is focused on.

- 3) Besides for methodologies and analyses, any possible thoughts or ideas about what factors might be influencing how multisensory events are integrated into time?

Minor Issues:

- 3) Paragraph indentation is inconsistent throughout the manuscript.

- Methods:

- 4) The first paragraph (lines 1 – 8) of the “Apparatus and stimuli” section has a lot of information that is repeated throughout the rest of the methods. It might be more appropriate later in the section or in the “Procedure” section.

- 5) Subheadings could be useful to organize and describe the different types of stimuli used.

- 6) For the 2nd criteria for data exclusion, how/what was used to determine if a curve had failed to converge on a solution for fitting a sigmoid function?

Reviewer #2: Smit and colleagues tested the effect of visual ‘compellingness’ on visual-tactile temporal integration windows. To this aim, they presented videos of hands (oriented in different directions) and a sponge being touched as visual stimuli. Using Bayes factors, the authors confirmed the absence of any effects on visual-tactile temporal order judgments. The study is carefully designed, results and analysis are sound. However, the authors might want to consider the alternative explanations for the absence of an effect listed below.

The measured JNDs a very large compared to previous studies (e.g., Keetels & Vroomen, 2008). This might be related to the nature of the visual stimuli; their onset could be harder to identify than that of simple flashes. Importantly, it is possible that the visual stimulus category had no influence on temporal integration windows because the noise of the visual temporal estimate dominated over the general assumption of a common cause in causal inference (Koerding et al., 2007, Eq. 2). In other words, the likelihoods of a common and of separate causes (Eqs. 4 and 6) might have been so small, that the size of p common hardly influenced the probability assigned to a common cause.

Some additional differences between the different studies should be included in the discussion. 1) Some of the studies used 3d hand stimuli, others 2d hand stimuli. 2) When a 2d hand is presented via the monitor (as in the current study) there is a mismatch between the real hand which is rotated orthogonal to gravity and the depicted hand which is rotated in the same direction as gravity.

There are fundamental differences between the current study design (and that of other cited studies) and the rubber hand illusion. Most importantly, the rubber hand illusion has to be induced by synchronous stroking of both hands. Therefore, the authors might focus stronger on those studies with a closer relation to the present study and reduce the references to the rubber hand illusion.

It seems to be an interesting coincidence that after collection of a dozen datasets the BF provides strong evidence in favor of a form effects. The authors might want to comment on this.

p. 17 + Figure 3 Please clarify a) whether the Cauchy prior which parameters have been tested is a prior on the size of the effect and b) how the directionality of the effect could be ignored when setting a prior on the effect size. Moreover, it seems more reasonable to assume an effect size based on visual-tactile studies rather than mixing visual-tactile and visual-proprioceptive studies.

The PSE provides no relevant information and could be dropped entirely.

Did participants receive feedback during the practice block?

Figure 2A Please include error bars

p. 20 ‘95%CI’ -> ‘95% CI’

6. PLOS authors have the option to publish the peer review history of their article (what does this mean?). If published, this will include your full peer review and any attached files.

Reviewer #1: No

Reviewer #2: No

---

## [Author Response · Author response to Decision Letter 0]

19 Aug 2019

Reviewer #1

- The focus of this research is centered around the question of how sensory cues modulate multisensory integration for body perception. In particular, this study used a visuo-tactile temporal order judgment (TOJ) task to test whether visual form and orientations cues could cause temporal integration. Comparisons were made between three conditions (hand in a plausible orientation; hand in an implausible orientation; sponge as a non-hand object) - The authors provide an extensive literature review that looked at various aspects of multisensory integration. Individual studies are described in quite a lot of detail but are used to relate and put the current study into context with previous findings. - Potential issues and limitations of previous studies were effectively identified and they aimed to remove some of these in the current study. - The findings showed that the three conditions did not differ, implying that visual form and orientation cues do not modulate temporal integration of visuo-tactile bodily signals. The authors focus the discussion on the conflicting findings in previous studies and conclude that the underlying mechanisms involved in multisensory integration appear to be different depending on the modalities used.

Response: We thank the reviewer for the constructive comments and respond to each below. 

Major Issues:

1) Since the introduction jumps right into a detailed review of the literature, it was a bit unclear what the current study was going to look at until the final paragraph. This made it harder to follow the logic of how and why the research questions fit into existing literature.

Response: We have now added a clear aim for our study at the end of the first paragraph of the introduction (p. 3) to make it apparent early on why our study is important and how it fits into the existing literature.

2) The discussion repeats a lot of what was said in the introduction about the limitations and how the studies differ. A lot of those details included in the introduction about each study could be moved to the discussion to provide more support for the claims and conclusions made. This would also help to streamline the introduction and make it easier to follow what the current study is focused on.

Response: We thank the reviewer for this suggestion. We have now included one succinct paragraph (p. 7) describing the most relevant differences between the studies (task differences, ecological validity and small samples) in the Introduction to provide clear motivation for our study design. We moved most of the details to the “Discussion” section and we agree with the reviewer that this adds strength to our conclusions. We also moved the section on audio-visual studies to the “Discussion” section. The introduction is now more streamlined and focused.

3) Besides for methodologies and analyses, any possible thoughts or ideas about what factors might be influencing how multisensory events are integrated into time?

Response: The major factors that can influence how multisensory events are integrated in time are the spatial and temporal proximity of the signals. These relate to some of the methodological issues we raised but we had not discussed it explicitly, so thank you for the question. We have added a final paragraph to the “Discussion” section (p. 29) on this.

“Thus overall, the influence of high-level contextual cues on temporal integration remains unclear. However, the major factors that are thought to modulate how multisensory events are integrated in time are the temporal and spatial proximity of signals [42, 64]. Studies show for example that increased spatial separation between cross-modal signals improves sensitivity for temporal order in TOJ tasks [65-67]. One explanation is that closeness in space indicates a single underlying cause, which increases intersensory binding and limits access to the temporal properties of individual signals. Alternatively, when there is a spatial gap between stimuli, participant could also rely on spatial position as an additional cue to establish which modality was presented first. This in turn could lead to smaller JNDs compared to when there are no or very small spatial discrepancies [65]. With respect to our results, the spatial distance between the viewed object and the hidden hand could be another potential cause for the discrepant findings (also see Keys et al. [27] for a similar discussion). It is important to note, however, that our spatial differences are less than half the distance of studies that found positive effects (Ide and Hidaka [25], 30 – 50 cm). Future research could systematically investigate the role of spatial distance on temporal effects and how these spatial cues might interact with contextual body cues.” 

Minor Issues:

3) Paragraph indentation is inconsistent throughout the manuscript.

Response: We thank the reviewer for the thoroughness in checking the formatting. We have checked and corrected inconsistent paragraph indentation.

Methods:

4) The first paragraph (lines 1 – 8) of the “Apparatus and stimuli” section has a lot of information that is repeated throughout the rest of the methods. It might be more appropriate later in the section or in the “Procedure” section.

Response: We moved the first paragraph from the “Apparatus and stimuli” section to the “Task and procedure” section (p. 14). We changed the heading from “Procedure” to “Task and procedure” and the heading “Apparatus and stimuli” to just “Stimuli”. We changed Fig 1 A and B (p. 11) around so it’s in line with the text (stimuli first and then procedure). We also carefully checked for repetition and consolidated these sections.

5) Subheadings could be useful to organize and describe the different types of stimuli used. 

Response: We thank the reviewer for the feedback. We added the subheadings “Visual stimulus” and “Tactile stimulus” to our “Methods” section (p.11). We also used better paragraph structure to organise this section. 

6) For the 2nd criteria for data exclusion, how/what was used to determine if a curve had failed to converge on a solution for fitting a sigmoid function?

Response: We determined this by examining each plot. In the excluded subjects there are clear straight lines instead of the expected convergence on sigmoid functions. To make this clear to the reader we added the following information in the “Method” section (p. 16):

“We excluded data sets for participants based on three criteria: (1) a JND larger than three standard deviations from the group mean in any condition; (2) any of the three curves (representing the three conditions) failing to converge on a solution for fitting a sigmoid function (in the excluded subjects there are clear straight lines instead of the expected convergence on sigmoid functions); or (3) incomplete data due to a technical error or failure to perform the task.” 

Reviewer #2 

Smit and colleagues tested the effect of visual ‘compellingness’ on visual-tactile temporal integration windows. To this aim, they presented videos of hands (oriented in different directions) and a sponge being touched as visual stimuli. Using Bayes factors, the authors confirmed the absence of any effects on visual-tactile temporal order judgments. The study is carefully designed, results and analysis are sound. However, the authors might want to consider the alternative explanations for the absence of an effect listed below. 

Response: We thank the reviewer for the helpful comments and respond to each below. 

1) The measured JNDs a very large compared to previous studies (e.g., Keetels & Vroomen, 2008). This might be related to the nature of the visual stimuli; their onset could be harder to identify than that of simple flashes. Importantly, it is possible that the visual stimulus category had no influence on temporal integration windows because the noise of the visual temporal estimate dominated over the general assumption of a common cause in causal inference (Koerding et al., 2007, Eq. 2). In other words, the likelihoods of a common and of separate causes (Eqs. 4 and 6) might have been so small, that the size of p common hardly influenced the probability assigned to a common cause. 

Response: We thank the reviewer for this point. If we understand correctly, the reviewer is suggesting that even if there is an effect of visual context, this may have been too small to detect due to a noisy inference caused by the visual touch stimulus. An important motivation for our study was to use naturalistic and complex stimuli to investigate temporal processes for body perception more broadly. Indeed our measured JNDs are relatively large compared to studies that involved more simple stimuli. This could be due to stimulus complexity as sensitivity for temporal order on a TOJ task has generally been shown to deteriorate when more complex stimuli are used (Vatakis & Spence, 2006; Vroomen & Keetels, 2010). This would explain why Maselli, Kilteni, López-Moliner, and Slater (2016) reported similarly large JNDs (up to 180ms) because their study also involved complex and moving visual stimuli (e.g., a rotating wheel that moved towards the participant’s virtual finger). However, it is possible that the noisiness of the data mean that an effect exists but that it’s too small in this context to pick up. We have added the following to the manuscript to acknowledge this (p. 25).

“Noise due to data variability may strongly affect outcomes on a TOJ task, especially when sample sizes are small [29, 30, 56]. To make sure our experiment produced sensitive data to either support our alternative or null hypothesis we based our sample size on a Bayesian analysis. Of course, it is always possible that there is an effect that is so small in the context of the variance within the task and participants, that we have not detected it. However, our methods give us considerable evidence for there being no effect of visual cues on visuo-tactile temporal integration of bodily signals.” 

2) Some additional differences between the different studies should be included in the discussion. 1) Some of the studies used 3d hand stimuli, others 2d hand stimuli. 2) When a 2d hand is presented via the monitor (as in the current study) there is a mismatch between the real hand which is rotated orthogonal to gravity and the depicted hand which is rotated in the same direction as gravity. 

Response: We thank the reviewer for this insightful point and agree that the difference in using 2D or 3D hand images and the orthogonal rotation of the hand are important to note. We have now included this in the paragraphs on ecological validity in our “Discussion” section (p.23), please see below.

“Another aspect that could have influenced the ecological validity of the visual stimulus is whether the visual hand stimulus is presented in 2D or 3D. Both Ide and Hidaka [25] and Maselli et al. [26] reported an effect with 2D and 3D stimuli respectively whereas the current study and Keys et al. [27] found evidence for the null with 2D and 3D hands. Further, previous studies show that simple 2D hand images are sufficient to establish interactions between visual and tactile stimuli [25, 54, 55]. Thus, it does not seem to be the dimensionality of the visual hand that determines the effect (or lack thereof). In addition, we presented the hand on a computer screen which results in a mismatch in the orthogonal rotation between the presented hand and the participant’s own hand. This could have influenced our results, however this seems improbable as Keys et al. [27] presented their hand stimuli in the same orthogonal rotation as the participant’s hand and also reported evidence for the null. Future studies could investigate any potential effect of the orthogonal rotation of the hand on temporal integration.”

3) There are fundamental differences between the current study design (and that of other cited studies) and the rubber hand illusion. Most importantly, the rubber hand illusion has to be induced by synchronous stroking of both hands. Therefore, the authors might focus stronger on those studies with a closer relation to the present study and reduce the references to the rubber hand illusion. 

Response: We thank the reviewer for this comment and it has prompted us to clarify why we discuss the rubber hand illusion. Our highlight of research with the rubber hand illusion was to demonstrate that besides spatial and temporal cues, visual form and orientation cues play an important role in the spatial integration of visual and proprioceptive inputs. This suggests that visual cues might also influence the integration of visual and tactile temporal inputs, which forms the motivation for our study. We have added a note to clarify this in the “Introduction” section (p. 3), please see below. “For instance, the RHI is reduced when the object lacks certain hand-like features [8], or when it is rotated at an improbable angle in relation to the participant’s own body [9, 10]. This demonstrates that besides temporal cues, visual form and orientation cues play an important role in perceiving one’s own body. However, the mechanism by which visual cues might modulate the integration of multisensory bodily stimuli still remains unclear. Further research into this process is fundamental for our understanding of how we perceive our bodies and interact with objects in the world around us. Motivated by the RHI literature, in this study, we use videos of touch combined with a felt touch to investigate if form and orientation cues directly influence the temporal integration of visual and tactile inputs.”

4) It seems to be an interesting coincidence that after collection of a dozen datasets the BF provides strong evidence in favor of a form effects. The authors might want to comment on this.”

Response: We agree with the reviewer and discuss this observation in the “Bayes factors and robustness check” section (p. 20). Our study really emphasises the need for large samples to be sure positive effects are robust.

5) p. 17 + Figure 3 Please clarify a) whether the Cauchy prior which parameters have been tested is a prior on the size of the effect and b) how the directionality of the effect could be ignored when setting a prior on the effect size. Moreover, it seems more reasonable to assume an effect size based on visual-tactile studies rather than mixing visual-tactile and visual-proprioceptive studies.

Response: We thank the reviewer for this point. We have now specified in the “Data analysis” section (p. 17) that we used the default Cauchy prior for the size of the effect and that we specified a non-directional alternative hypothesis. Based on our review of the literature, there could have been an effect in either direction and thus our alternative hypothesis was not directional (as pre-registered). Based on the reviewer’s feedback, we have re-calculated the Bayes factor with the one-tailed Dienes calculator – we found the following evidence for the null: BF = 5.08 (orientation) and BF = 4.18 (form). This evidence is slightly weaker than we reported in the paper (evidence for the null, BF= 6.94 for orientation and BF = 5.61 for form) but still provides moderate evidence. As previous evidence suggested there could be effects in either direction, we would prefer to stick with our pre-registered non-directional prior, but if the reviewer feels strongly we could add an additional comment about this post-hoc check.

The reason we mixed evidence from visuo-tactile and visuo-proprioceptive studies to establish our predicted effect size is because it gave us more data and hence a better informed prior. However, if we look only at the visuo-tactile studies that found an effect, we get the same predicted effect size: Maselli et al. (2016) reported 28 ms and Ide and Hidaka (2013) reported 12 ms; average = 20 ms. Again, if the reviewer would like, we can add an additional comment to this effect to the paper.

6) The PSE provides no relevant information and could be dropped entirely.

Response: We have cut the PSS report and analysis. Instead, in the “Data analysis” section (p. 15) we refer to our PSS data on the project’s OSF page in case the reader would like to find this information. 

7) Did participants receive feedback during the practice block?

Response: Yes, participants received feedback during the practice block and we have now added this information to the “Procedure” section (p. 14). 

8) Figure 2A Please include error bars p. 20 ‘95%CI’ -> ‘95% CI’

Response: We have now included 95% CI in this figure (p. 19).

References

Costantini, M., & Haggard, P. (2007). The rubber hand illusion: sensitivity and reference frame for body ownership. Consciousness and cognition, 16(2), 229-240.

Ehrsson, H. H., Spence, C., & Passingham, R. E. (2004). That's my hand! Activity in premotor cortex reflects feeling of ownership of a limb. Science, 305(5685), 875-877.

Ide, M., & Hidaka, S. (2013). Visual presentation of hand image modulates visuo–tactile temporal order judgment. Experimental Brain Research, 228(1), 43-50.

Igarashi, Y., Kitagawa, N., & Ichihara, S. (2004). Vision of a pictorial hand modulates visual-tactile interactions. Cognitive, Affective, & Behavioral Neuroscience, 4(2), 182-192.

Igarashi, Y., Kitagawa, N., Spence, C., & Ichihara, S. (2007). Assessing the influence of schematic drawings of body parts on tactile discrimination performance using the crossmodal congruency task. Acta psychologica, 124(2), 190-208.

Keys, R. T., Rich, A. N., & Zopf, R. (2018). Multisensory temporal processing in own-body contexts: plausibility of hand ownership does not improve visuo-tactile asynchrony detection. Experimental Brain Research, 236(5), 1431-1443.

Maselli, A., Kilteni, K., López-Moliner, J., & Slater, M. (2016). The sense of body ownership relaxes temporal constraints for multisensory integration. Scientific reports, 6, 30628.

Radeau, M. (1994). Auditory-visual spatial interaction and modularity. Cahiers de Psychologie Cognitive/Current Psychology of Cognition, 13(1), 3–51.

Shimada, S., Fukuda, K., & Hiraki, K. (2009). Rubber hand illusion under delayed visual feedback. PloS one, 4(7), e6185.

Spence, C., Baddeley, R., Zampini, M., James, R., & Shore, D. I. (2003). Multisensory temporal order judgments: When two locations are better than one. Perception & Psychophysics, 65(2), 318-328.

Spence, C., Shore, D. I., & Klein, R. M. (2001). Multisensory prior entry. Journal of Experimental Psychology: General, 130(4), 799.

Tsakiris, M., Carpenter, L., James, D., & Fotopoulou, A. (2010). Hands only illusion: multisensory integration elicits sense of ownership for body parts but not for non-corporeal objects. Experimental Brain Research, 204(3), 343-352.

Vatakis, A., & Spence, C. (2006). Audiovisual synchrony perception for music, speech, and object actions. Brain research, 1111(1), 134-142.

Vroomen, J., & Keetels, M. (2010). Perception of intersensory synchrony: a tutorial review. Attention, Perception, & Psychophysics, 72(4), 871-884.

Zampini, M., Shore, D. I., & Spence, C. (2003). Audiovisual temporal order judgments. Experimental Brain Research, 152(2), 198-210.

---

## [Decision Letter · Decision Letter 1]

17 Sep 2019

PONE-D-19-14549R1

Visual body form and orientation cues do not modulate visuo-tactile temporal integration

PLOS ONE

Dear Dr Smit,

Thank you for submitting your revised paper to PLOS ONE. As you will see, both reviewers were generally satisfied with the revisions you have made based on their previous comments. Reviewer 2, however, continues to have concerns about your interpretation of your results, and has made some details comments. I would therefore like to invite you to respond to this issue, and to review your paper appropriately. Based on your response, I will make a decision about whether to send the paper back to Reviewer 2 for further comment.

We would appreciate receiving your revised manuscript by Nov 01 2019 11:59PM. To enhance the reproducibility of your results, we recommend that if applicable you deposit your laboratory protocols in protocols.io, where a protocol can be assigned its own identifier (DOI) such that it can be cited independently in the future. For instructions see: http://journals.plos.org/plosone/s/submission-guidelines#loc-laboratory-protocols

We look forward to receiving your revised manuscript.

Kind regards,

Matthew Longo, Ph.D.

Academic Editor

PLOS ONE

Reviewers' comments:

Reviewer's Responses to Questions

**Comments to the Author**

1. If the authors have adequately addressed your comments raised in a previous round of review and you feel that this manuscript is now acceptable for publication, you may indicate that here to bypass the “Comments to the Author” section, enter your conflict of interest statement in the “Confidential to Editor” section, and submit your "Accept" recommendation.

Reviewer #1: All comments have been addressed

Reviewer #2: (No Response)

2. Is the manuscript technically sound, and do the data support the conclusions?

Reviewer #1: Yes

Reviewer #2: Partly

3. Has the statistical analysis been performed appropriately and rigorously? 

Reviewer #1: Yes

Reviewer #2: Yes

4. Have the authors made all data underlying the findings in their manuscript fully available?

Reviewer #1: Yes

Reviewer #2: Yes

5. Is the manuscript presented in an intelligible fashion and written in standard English?

Reviewer #1: Yes

Reviewer #2: Yes

6. Review Comments to the Author

Reviewer #1: Great job with responding to and addressing all of the comments. The changes have greatly improved the introduction and overall paper. I have no additional comments or concerns.

Reviewer #2: The authors did a nice job in revising the manuscript. However, I must insist that there is a good explanation of the results within the causal inference framework and, thus, the conclusion cannot include a rejection of or any speculation about the causal inference model. My previous comment on this issue obviously did not a good job in explaining why this is the case, so I have included more details about the model logic.

Bayes in the Brain

Rejecting a model without fitting it to the data is a problematic endeavor, as it is very hard to predict the outcome of complex models without model implementation.

The current study shows that visual form and orientation cues have no effect on visual-tactile temporal integration windows. However, this result is not sufficient to reject the causal inference model for temporal integration.

The reason lies in the inner workings of the causal inference model. The authors assume that the visual manipulations affect participants’ common cause prior which in turn would affect the weight given to the integrated estimate (also see comment below about terminology). Such an effect could become visible in the size of the integration window, i.e., the JND.

However, the size of the mixture weights depends not exclusively on the common cause prior but also on the sensory signals, i.e., the likelihoods and the sensory prior(s): In the model, the probability that the two measurements that arrived in the brain stem from one cause (P(x_t, x_v|C=1)) and the probability that the same signals originate from different causes are calculated (P(x_t, x_v|C=2)) and then multiplied with the common cause prior to derive a posterior estimate of the probability of a common cause given the signals (P(C=1|x_t, x_v)). This probability is then used to calculate the mixture weights. As a consequence, a change in the common cause prior will only measurably affect the mixture weights and thus the integration window, if the sensory information about a common cause is ambiguous.

Here, it seems very unlikely that this is the case. The visual stimuli are likely to result in wide likelihoods (which has nothing to do with sample size, that is a different kind of noise) because the exact time point of the stimulation might be hard to extract compared to a flash of light. The large JNDs indeed could indicate such wide likelihoods. If the sensory information of one modality is quite uncertain, the probability that the two measurements that arrived in the brain stem from one cause (P(x_t, x_v|C=1)) will be high. Thus, the influence of changes in the common cause prior will not be noticeable.

Additionally, the large JNDs might indicate a very high common cause prior in any condition, which would make it even more unlikely to notice the effects of a change in the common cause prior with visual form and orientation. This is because an increase in the common cause prior from 0.4 to 0.6 could lead to more drastic behavioral changes than an increase from 0.7 to 0.9 (and again, this depends on the sensory information).

Taken together, the authors cannot make any claim about the validity of the causal inference model for temporal integration. However, this is not necessarily bad. The study used realistic stimuli. Thus, even within the model we learn that under naturalistic conditions any effect of general information regarding the unity of the signals (i.e., the common cause prior) diminishes due to the dominance of sensory information.

Side note: as laid out above, in the model, the brain does not decide between integration and separation (as the text sometimes states) but derives both estimates and integrates them weighted by the probability of the underlying scenario. There are now multiple modeling and imaging studies that support that both estimates are derived.

From a model perspective, integration itself can either be based on the optimal weights or not, but integration cannot be weak or strong (the text speaks of strong temporal integration). The same holds from a physiological perspective, the effects can be weak or strong but integration itself can only be present or absent. The multisensory community usually uses the terms wide and narrow temporal integration windows.

Bayesian Statistics

Some aspects of the Bayesian analysis are still a bit confusing, which might be due to the fact that toolboxes with built-in options were used. I recommend simplifying, so that readers can concentrate on the main message.

1) The visual-proprioceptive studies should not be used for the effect size estimation, simply because touch and proprioception are different modalities.

2) Given the small difference between the BF scores, the bi-directional hypothesis should be fine.

3) Figure 3 might well do more harm than good by simply distracting readers. Readers who are not informed about Bayesian statistics will not understand why the choice of prior over effect size matters. Readers savvy in Bayesian statistics usually are not too fond of toolboxes and thus will not really care for Figure 3. Those readers who need to be convinced that the effect which is looks very evident in Figure 2 is not dependent on the choice of prior should be satisfied with the text saying that the robustness was verified.

Minor points

- The results figures look very blurry in the reviewer pdf. The authors might want to check what is going on there before the paper goes into production.

- There are several instances in which 95%CI should be replaced with 95% CI.

7. PLOS authors have the option to publish the peer review history of their article (what does this mean?). If published, this will include your full peer review and any attached files.

Reviewer #1: No

Reviewer #2: No

---

## [Author Response · Author response to Decision Letter 1]

4 Oct 2019

Response to Reviewer #2 

(Reviewer)

The authors did a nice job in revising the manuscript. However, I must insist that there is a good explanation of the results within the causal inference framework and, thus, the conclusion cannot include a rejection of or any speculation about the causal inference model. My previous comment on this issue obviously did not a good job in explaining why this is the case, so I have included more details about the model logic. 

Bayes in the Brain 

Rejecting a model without fitting it to the data is a problematic endeavor, as it is very hard to predict the outcome of complex models without model implementation. The current study shows that visual form and orientation cues have no effect on visual-tactile temporal integration windows. However, this result is not sufficient to reject the causal inference model for temporal integration. The reason lies in the inner workings of the causal inference model. The authors assume that the visual manipulations affect participants’ common cause prior which in turn would affect the weight given to the integrated estimate (also see comment below about terminology). Such an effect could become visible in the size of the integration window, i.e., the JND. However, the size of the mixture weights depends not exclusively on the common cause prior but also on the sensory signals, i.e., the likelihoods and the sensory prior(s): In the model, the probability that the two measurements that arrived in the brain stem from one cause (P(x_t, x_v|C=1)) and the probability that the same signals originate from different causes are calculated (P(x_t, x_v|C=2)) and then multiplied with the common cause prior to derive a posterior estimate of the probability of a common cause given the signals (P(C=1|x_t, x_v)). This probability is then used to calculate the mixture weights. As a consequence, a change in the common cause prior will only measurably affect the mixture weights and thus the integration window, if the sensory information about a common cause is ambiguous. Here, it seems very unlikely that this is the case. The visual stimuli are likely to result in wide likelihoods (which has nothing to do with sample size, that is a different kind of noise) because the exact time point of the stimulation might be hard to extract compared to a flash of light. The large JNDs indeed could indicate such wide likelihoods. If the sensory information of one modality is quite uncertain, the probability that the two measurements that arrived in the brain stem from one cause (P(x_t, x_v|C=1)) will be high. Thus, the influence of changes in the common cause prior will not be noticeable.

Additionally, the large JNDs might indicate a very high common cause prior in any condition, which would make it even more unlikely to notice the effects of a change in the common cause prior with visual form and orientation. This is because an increase in the common cause prior from 0.4 to 0.6 could lead to more drastic behavioral changes than an increase from 0.7 to 0.9 (and again, this depends on the sensory information). Taken together, the authors cannot make any claim about the validity of the causal inference model for temporal integration. However, this is not necessarily bad. The study used realistic stimuli. Thus, even within the model we learn that under naturalistic conditions any effect of general information regarding the unity of the signals (i.e., the common cause prior) diminishes due to the dominance of sensory information. 

Response: We thank the reviewer for providing more detail in relation to the previous comment. We have added an additional paragraph in the discussion section to address the relatively large JNDs and the possibility that any additional influence of the visual manipulations on the the assumption of a common cause might have been too small due to noise from the visual stimulus, and hence potentially not measurable in the context of the stimuli we used:

“The measured JNDs in the current study are relatively large compared to previous studies that used more basic stimuli (e.g., Keetels and Vroomen [1]). This could be due to the complex nature of the visual stimuli as it might be harder to extract their onset compared to simple stimuli such as light flashes. Sensitivity for temporal order on a TOJ task has generally been shown to deteriorate when more complex stimuli are used [2, 3]. This sensory temporal noise in the visual touch signal could influence the probability estimate for a common cause for the visual and tactile touch signals [4]. It is thus possible that any additional influence from the visual cues (form and orientation) on the the assumption of a common cause (i.e., common cause prior) might have been too small and not measurable in the context of the stimuli we used. In other words, a Bayesian causal inference mechanism for temporal integration which takes into account body form and orientation could in principle still hold up and further testing and computational modeling would be required to evaluate the evidence for this model. However, an important motivation for our study was to use naturalistic stimuli to investigate temporal processes for body perception and our results provide evidence that under these conditions, the relative degree of temporal integration is not modulated by the body form and orientation context.” (p. 28)

Further, as noted by the reviewer, we did not fit our data to the model and therefore cannot refute it. We have changed the wording in the abstract and conclusion so that it only states that our results are not in line with broad predictions by the model. We also mention the need for future computational modelling:

“Results show that visual cues do not modulate visuo-tactile temporal order judgements. This is not in line with the idea that bodily signals indicating oneself influence the integration of multisensory signals in the temporal domain.” (p.2)

“Conversely, the current and previous findings suggest that a plausible context does not result in an increased degree of temporal integration of bodily signals and hence are inconsistent with the idea that cues indicating oneself modulate the integration of multisensory signals in the temporal domain. Further, research into the effects of higher-level cues such as context on temporal integration is limited and conflicting in the multisensory literature more broadly. Thus generally, more research, including computational modelling, is needed to provide better insight into the factors that influence how multisensory events are integrated.” (p.31)

(Reviewer)

Side note: as laid out above, in the model, the brain does not decide between integration and separation (as the text sometimes states) but derives both estimates and integrates them weighted by the probability of the underlying scenario. There are now multiple modeling and imaging studies that support that both estimates are derived. From a model perspective, integration itself can either be based on the optimal weights or not, but integration cannot be weak or strong (the text speaks of strong temporal integration). The same holds from a physiological perspective, the effects can be weak or strong but integration itself can only be present or absent. The multisensory community usually uses the terms wide and narrow temporal integration windows. 

Response: We thank the reviewer for this point and the opportunity to clarify this aspect in the manuscript. We have revised the section where we introduce and discuss the Bayesian causal inference model so that it more accurately reflects the relative weighting of causal structures for common and separate causes and thus the relative degree of integration:

“The brain constantly receives signals from different sensory modalities with some variability between the exact timing or location, and these are either combined into the same multisensory event or kept separate [5, 6]. One proposal is that the processing of multisensory signals relies on computational mechanisms for causal inference to determine the probability that the individual unisensory signals belong to the same object or event. This account holds that the brain computes probabilities for common and separate causes, which then provide the weights given to the integrated and separated perceptual estimates. These relative weights determine the degree of integration versus separation of multisensory signals [7]. Bayesian causal inference models can therefore provide a unified theory for the perception of multisensory events, including their spatial and temporal characteristics [4, 8-11]. On this view, the degree of integration versus separation can be influenced by previous knowledge that signals belong to one and the same object or event, and repeated experience that signals are statistically likely to co-occur [12-14]. (p.3)

“A causal inference process might also govern the binding and integration of bodily signals [15]. Depending on whether the brain infers a common cause for inputs or not, it integrates or segregates spatial and temporal signals coming from visual, tactile and proprioceptive modalities. Visual cues such as body form and orientation could function as causal binding factors (i.e., influence the relative probabilities for a common versus a separate cause) as these indicate whether or not inputs originated from the same source (e.g., one’s own hand).” (p.4)

“A Bayesian causal inference model proposes that multisensory perception relies on causal inference to establish the probabilities as to which unimodal inputs share a common source and should therefore be integrated (or otherwise segregated due to separate sources).” (p.21)

We have also changed some of the wording throughout the text which is highlighted in yellow. For example, we avoid the terms “weaker versus stronger” integration and instead use “the relative degree of integration” (see for example Shams and Beierholm [7] who also use these terms).

For overall clarity however, we decided not to use the terminology ‘temporal integration window / temporal binding window’ (often defined as “the epoch of time within which stimuli from different modalities are likely to be integrated and perceptually bound”, e.g., Wallace and Stevenson [16]). This integration window can be measured by manipulating the temporal difference between multisensory stimuli and then test how this affects perceptual integration (see for example the McGurk effect: Van Wassenhove, Grant and Poeppel [17] or the Rubber Hand Illusion: Shimada, Fukuda and Hiraki [18]). Instead we predominantly talk about larger / smaller JNDs as we tested the influence of context on the relative degree of multisensory temporal integration (i.e., the threshold for asynchrony at which temporal order can be reliably established). It is possible that even when signals are perceived as temporally separate due to temporal asynchronies, these signals might still be perceptually bound on non-temporal dimensions, as for example observed in the rubber hand illusion. We have mentioned this in the Discussion:

“This suggests that even when signals are processed as being temporally separate, in bodily illusions these might still be perceptually integrated on a non-temporal dimension. Perception in bodily illusions is likely influenced not just by temporal cues but also by other types of information such as the plausibility of the visual context and factors such as temporal recalibration or attention [6, 43]. This may result in the observation of less strict temporal constraints as reported by Shimada et al. [5]. Our finding suggests that the visual context on its own does not modulate the degree of temporal integration.” (p. 28) 

(Reviewer)

Some aspects of the Bayesian analysis are still a bit confusing, which might be due to the fact that toolboxes with built-in options were used. I recommend simplifying, so that readers can concentrate on the main message.

Response: We have now simplified our “Bayesian analysis” section (see also our response under point 3 below).

(Reviewer)

1) The visual-proprioceptive studies should not be used for the effect size estimation, simply because touch and proprioception are different modalities. 

Response: We have made a note in the text based on the reviewer’s feedback. We no longer use visuo-proprioceptive studies to estimate the predicted effect size:

“We preregistered this study including the two hypotheses, planned methods and the data analysis plan before data collection. We followed this plan with two exceptions. First, we listed the subject pool as undergraduate students but also tested participants from the university community who responded to university advertising (e.g., postgraduate students). Second, for our Bayesian analyses we specified a predicted effect size based on previous studies including those that looked at the effect of visual cues on visuo-proprioceptive temporal integration. Based on comments from a reviewer, we subsequently changed the effect size calculation to only include studies that investigated visuo-tactile integration specifically. This does not change the predicted effect size (20 ms), so the actual results did not change due to this diversion.” (p.9)

(Reviewer)

2) Given the small difference between the BF scores, the bi-directional hypothesis should be fine. 

Response: Thank you.

(Reviewer)

3) Figure 3 might well do more harm than good by simply distracting readers. Readers who are not informed about Bayesian statistics will not understand why the choice of prior over effect size matters. Readers savvy in Bayesian statistics usually are not too fond of toolboxes and thus will not really care for Figure 3. Those readers who need to be convinced that the effect which is looks very evident in Figure 2 is not dependent on the choice of prior should be satisfied with the text saying that the robustness was verified. 

Response: For Figure 3 we have taken out sections B and C so that it only shows the sequential plotting of Bayes factors plotted in R based on our informed prior. We have also adapted this in the text and only briefly mention that robustness was verified in JASP: 

“To check the impact of the prior on our analysis, we also performed an analysis and robustness check in JASP using the default priors. This indicated that evidence for the null hypothesis is stable across a range of specified parameters which suggests that our analysis is robust.” (p.20)

(Reviewer)

Minor points 

The results figures look very blurry in the reviewer pdf. The authors might want to check what is going on there before the paper goes into production. 

Response: We thank the reviewer for checking the quality of the figures. We agree the figures in the compiled PDF look somewhat blurry. We have checked the uploaded figures by downloading them with the links found at the top of page 48, 49 and 50 of the compiled PDF and these do appear to have high resolution. We hope the publisher will let us know if we could provide the images in a different format to improve the compiled PDF.

(Reviewer)

There are several instances in which 95%CI should be replaced with 95% CI. 

Response: We have amended this in the text.

References

1. Keetels M, Vroomen J. Temporal recalibration to tactile–visual asynchronous stimuli. Neuroscience letters. 2008;430(2):130-4.

2. Vatakis A, Spence C. Audiovisual synchrony perception for music, speech, and object actions. Brain research. 2006;1111(1):134-42.

3. Maselli A, Kilteni K, López-Moliner J, Slater M. The sense of body ownership relaxes temporal constraints for multisensory integration. Scientific Reports. 2016;6:30628.

4. Körding KP, Beierholm U, Ma WJ, Quartz S, Tenenbaum JB, Shams L. Causal inference in multisensory perception. PLoS one. 2007;2(9):e943.

5. Calvert G, Spence C, Stein BE. The handbook of multisensory processes. Cambridge: MIT press; 2004.

6. Stein BE, Meredith MA. The merging of the senses. Cambridge: MIT Press; 1993.

7. Shams L, Beierholm UR. Causal inference in perception. Trends in Cognitive Sciences. 2010;14(9):425-32.

8. Beierholm UR, Quartz SR, Shams L. Bayesian priors are encoded independently from likelihoods in human multisensory perception. Journal of Vision. 2009;9(5):23-.

8

9. Wozny DR, Beierholm UR, Shams L. Probability matching as a computational strategy used in perception. PLoS computational biology. 2010;6(8):e1000871.

10. Wozny DR, Beierholm UR, Shams L. Human trimodal perception follows optimal statistical inference. Journal of Vision. 2008;8(3):24-.

11. Shams L, Ma WJ, Beierholm U. Sound-induced flash illusion as an optimal percept. Neuroreport. 2005;16(17):1923-7.

12. Bresciani J-P, Dammeier F, Ernst MO. Vision and touch are automatically integrated for the perception of sequences of events. Journal of Vision. 2006;6(5):2-.

13. Helbig HB, Ernst MO. Knowledge about a common source can promote visual—haptic integration. Perception. 2007;36(10):1523-33.

14. Ernst MO. Learning to integrate arbitrary signals from vision and touch. Journal of Vision. 2007;7(5):7-.

15. Samad M, Chung AJ, Shams L. Perception of body ownership is driven by Bayesian sensory inference. PloS one. 2015;10(2):e0117178.

16. Wallace MT, Stevenson RA. The construct of the multisensory temporal binding window and its dysregulation in developmental disabilities. Neuropsychologia. 2014;64:105-23.

17. Van Wassenhove V, Grant KW, Poeppel D. Temporal window of integration in auditory-visual speech perception. Neuropsychologia. 2007;45(3):598-607.

18. Shimada S, Fukuda K, Hiraki K. Rubber hand illusion under delayed visual feedback. PloS one. 2009;4(7):e6185.

---

## [Editor Report · Decision Letter 2]

8 Oct 2019

Visual body form and orientation cues do not modulate visuo-tactile temporal integration

PONE-D-19-14549R2

Dear Dr Smit,

We are pleased to inform you that your manuscript has been judged scientifically suitable for publication and will be formally accepted for publication once it complies with all outstanding technical requirements.

With kind regards,

Matthew Longo, Ph.D.

Academic Editor

PLOS ONE
---

## [Editor Report · Acceptance letter]

16 Oct 2019

PONE-D-19-14549R2 

Visual body form and orientation cues do not modulate visuo-tactile temporal integration 

Dear Dr. Smit:

I am pleased to inform you that your manuscript has been deemed suitable for publication in PLOS ONE. Congratulations! Your manuscript is now with our production department. 

With kind regards,

on behalf of

Dr Matthew Longo 

Academic Editor

PLOS ONE